# Preferences for Indoor Environmental and Social Comfort of Outpatient Staff during the COVID-19 Pandemic, an Explanatory Study

**DOI:** 10.3390/ijerph18147353

**Published:** 2021-07-09

**Authors:** AnneMarie Eijkelenboom, Marco A. Ortiz, Philomena M. Bluyssen

**Affiliations:** Faculty of Architecture and the Built Environment, Delft University of Technology, 2628BL Delft, The Netherlands; m.a.ortizsanchez@tudelft.nl (M.A.O.); P.M.Bluyssen@tudelft.nl (P.M.B.)

**Keywords:** hospitals, COVID-19, preferences, outpatient areas, multisensory, IEQ, semi-structured interviews, design, health, hospital staff

## Abstract

While the pressure on hospital workers keeps growing, they are generally more dissatisfied with their comfort than other occupants in hospitals or offices. To better understand the comfort of outpatient workers in hospitals, clusters for preferences and perceptions of the indoor environmental quality (IEQ) and social comfort were identified in a previous study before the outbreak of the coronavirus disease 2019 (COVID-19) pandemic. This qualitative study explains the outpatient workers’ main preferences for comfort during the COVID-19 pandemic. Semi-structured interviews and photo-elicitation were used. Contextual changes due to the COVID-19 pandemic were included. The questions in the interviews were based on the characteristics of the profiles, corresponding with the clusters. The data were analyzed with content analysis according to the steps defined by Gioia. Seventeen outpatient workers who had been part of the previous study participated. For some outpatient workers differentiation of preferences was illogical due to interrelations and equal importance of the comfort aspects. The main changes in perceptions of comfort due to the pandemic were worries about the indoor air quality and impoverished interaction. Because the occupants’ preferences for comfort can change over time, it was suggested that further development of occupant profiles needs to accommodate changes.

## 1. Introduction

As the pressure on hospital workers grows due to the increasing demand for healthcare [1], in the Netherlands almost half of these hospital workers experienced high work pressure in 2019 [2]. Stress can have a negative influence on work satisfaction, performance, and healthcare staff turnover [3,4,5]. One of the factors that can influence stress at work is the physical environment [6]. For example, previous studies showed that work dissatisfaction and hospital workers’ turnover were positively related to dissatisfaction with noise [7] and the length of daylight exposure [8].

The study of comfort, specifically that of hospital workers, is important because previous studies identified a tendency of higher dissatisfaction of hospital workers with the comfort compared to patients [9,10,11]. Other studies have shown that hospital staff are also less satisfied with their comfort and suffer more from building-related symptoms than occupants in office buildings [12,13]. To improve the health and comfort of the hospital workers it is important to better understand their needs and preferences.

### 1.1. Background

It has been suggested by several authors that IEQ as well as social comfort aspects are important to understand health and comfort [14,15,16]. Privacy and interaction have been included in previous studies. For example, nurses that moved from open bay wards to a ward with 100% single patient rooms, missed the informal interaction with colleagues in the new wards [17]. Also, the exchange of medical information was better in open wards, that comprised of 36 beds without separation walls, than in bay wards with walls between 4–6 beds [18]. Another example is the relation of the type of communication (case related or comforting) with room types in an emergency department [19].

As hospitals are complex buildings that accommodate a large variety of activities, the hospital workers’ needs for comfort can vary between hospital departments [20,21]. Therefore, it is important to gain insights into comfort in specific hospital areas, such as outpatient areas. Because outpatient areas have been understudied, a survey on the comfort and health of hospital workers in these areas was performed in the spring of 2019 [22]. In these areas hospital workers consult, diagnose, and treat patients who do not stay overnight in the hospital. Data from 556 outpatient workers were collected with a questionnaire and data from 107 rooms, the building services, and locations of six hospitals were collected with a building inspection. The questionnaire included the occupants’ assessment of IEQ factors, (indoor air quality, thermal comfort, lighting quality, and acoustics) and social comfort aspects (privacy, distraction, interaction). The study identified that the satisfaction with IEQ and social comfort varied, depending on the room types where the outpatient workers worked most frequently. For example, those who worked more frequently in a consultation room were more satisfied with privacy than those who worked in offices.

Questions on the preferences that were closely related to the questions on the perception of IEQ and social comfort were also included, in order to identify differences in needs between individual occupants. Because comfort can vary between individuals, due to differences in reaction and sensitivity to building characteristics, profiles of outpatient workers were produced to justify the variation in comfort between individuals [23]. The profiles were produced with) Two-Step Cluster analysis in IBM SPSS Statistics 25. This is a segmentation method that is suitable to identify groups with similar needs [24]. Two sets of clusters were produced: one set including satisfaction and preferences for IEQ and another one including satisfaction and preferences for social comfort [25]. In both sets of clusters, the preferences had higher importance than dissatisfaction.

The strong differentiation of the preferences between the clusters brought in the question of how occupants differentiate their preferences. Previous studies identified that the preferences of hospital workers were associated with personal and work-related aspects, such as gender, age, working hours [26], working years, and the function of areas [27]. However, these studies did not explain how the occupants perceived their preferences and comfort. Studies on preferences for thermal comfort in offices [28,29] and housing [30,31] showed that needs, behavioural strategies, and control of equipment were interrelated with the specific context. Therefore, it is important to explain the preferences of occupants within their context.

However, the context of outpatient areas may have changed because of the outbreak and worldwide spread of the SARS-CoV-2 virus, leading to the coronavirus disease 2019 (COVID-19) pandemic in the spring of 2020 [32]. Previous studies showed that the outbreak of a serious infectious disease may increase work stress and affect the health of hospital workers [33,34,35,36]. As rapid changes in care processes may influence the perception of comfort, assessing the hospital environment is needed when changes occur [37].

### 1.2. Objective

Because of limited information on the hospital workers’ preferences for comfort, this study aims to explain the differences in preferences of outpatient workers that were identified in the clusters of the 2019 survey. This study provides insights into the comfort experience of outpatient workers during the COVID-19 pandemic and changes of their preferences during the pandemic.

The present study is part of a larger research project that aims to gain a better understanding of the health and comfort of outpatient workers. The study acts as a follow-up to the survey that was performed with outpatient workers of three hospital organizations in the Netherlands before the COVID-19 pandemic started [22,25] (Figure 1).

## 2. Materials and Methods

### 2.1. Study Design

A qualitative approach was selected because of the unknown context due to the COVID-19 pandemic and limited information on the experience of comfort preferences. A qualitative approach allows to study possible changes over time and to listen to the real-life experiences of occupants in the built environment [38]. Qualitative data are suitable to refine and explain the results of the quantitative analysis [39] that was used to produce the clusters and their profiles. This explanatory study was based on a pragmatic worldview, considering both the physical world and human experience [40].

The study design combined photography and semi-structured interviews to gain an in-depth understanding of the preferences of the outpatient workers. The critical appraisal skills programme checklist for qualitative research was used for the study design [41].

The participants were selected out of the respondents who participated in the previous survey on comfort and health [25] and had shared their e-mail addresses to participate in a follow-up study. The study intended to include outpatient workers with different functions from all three hospital organizations and all IEQ and social clusters, to gain a representative overview. Participants who were relocated to a location that was not part of the survey or who could not work at one of the six locations due to illness, were not included.

Invitations and reminders were sent by e-mail in October 2020. The outpatient workers who accepted the invitation to participate received a proposal for dates and times and an instruction sheet to take the photographs (Figure 2). In the instruction sheet, it was explained that the photographs were needed to provide insights into the importance of IEQ and social comfort aspects during the interview. It was instructed that other rooms could be photographed to explain the importance of comfort aspects. IEQ was defined as: “aspects from the indoor environment: ventilation, temperature, noise and light”. Social comfort was defined as: “contact with others, distraction, safety, privacy, crowding”. To protect the privacy of patients and staff, instructions were given to exclude persons and personal information from the images.

Photographs were used because images support communication, as shown in studies on the perception of building characteristics [42]. Images in research have been used and tested before and support the narrative of real-life experiences [43]. For example, in a study that explored the patients’ experience when they were lying in a bed, patients took photographs in advance of the semi-structured interviews [44]. It was concluded that visual material cannot only illustrate visual but also auditory or other sensory qualities of the physical environment during the interviews. A practical reason to use photographs was to gain insights into the building characteristics while avoiding infection risk by the interviewer by visiting the hospitals. It should be noted that the interviewer had systematically inspected all hospital buildings before the pandemic started [25].

The semi-structured interviews were conducted via video calls or telephone calls, depending on the participant’s preference and technological possibilities. All interviews were audiotaped with Microsoft Teams with the consent of the participants. Semi-structured interviews were used because these enable to explore perceptions of the respondents, while they allow for differences in education, experience, and personal background between individuals [45]. The structure of the interviews enabled follow-up questions to be asked [46]. To reduce the potential bias of the researcher, the interviews started with an introduction and general questions about work. In the introduction it was explained that there was no right or wrong answer. The questions were phrased neutrally, and leading questions were avoided.

The interview guide comprised of newly developed questions, that were discussed and tested in a pilot. The interviews consisted of five main topics: work-related aspects, changes due to the COVID-19 pandemic, preferences for IEQ, preferences for social comfort, and ranking of preferences (Table 1). The subtopics were aspects that were differentiated by the profiles of the outpatient workers in the previous survey on comfort and health [25]. For example, the number of persons in the room was a subtopic because it varied between the clusters. A comparison with the data from the survey was done to check whether the room type, department, or location of the participants had been changed. The subtopic ‘Logic to differentiate preferences’ was added to gain insight into whether the participants experience clear differences between their preferences. The ranking was performed separately for comfort with IEQ-aspects and social comfort. The aspects that were identified in the previous survey as most important were used for the ranking. The IEQ-aspects were control of ventilation, sufficient fresh air, control of temperature, not too cold or hot, sufficient daylight, no annoyance by noise. For social comfort, the aspects were contact with patients and colleagues, a safe workplace, sufficient patient privacy, no distraction from noise. The structure of the interview was tested in a pilot interview with an outpatient worker from a similar hospital organization (teaching hospital). The pilot provided insights into the structure of the interview and the use of photographs. Consequently, the researcher piloted two interviews with two outpatient workers from similar hospital organizations.

### 2.2. Ethical Approval

The Ethics committee of the Delft University of Technology approved the study design on 5 October 2018. Data security was assessed by a data manager of the university. The data were stored on a secured server. Participation of the hospital organizations and participants was voluntary. Participants could participate only after their approval of informed consent. The letter of consent and procedure were discussed and approved with the project leader of each participating hospital organization in advance. If the participants had shared their e-mail address in the previous survey, it was separated from the dataset and secured in a separate document. Comparison of the individuals’ data between the survey and this follow-up study was enabled by a unique number that was assigned to each participant. To respect the privacy of the participants, persons are not traceable from the results presented.

### 2.3. Data Analysis

Data analysis was performed during the two phases, and in the steps shown in Figure 3. In the first phase inductive analysis was performed to structure the changes due to the COVID-19 pandemic. An inductive approach was used, because of the unknown context. In the second phase deductive analysis was performed to structure the main occupants’ preferences and enable comparison with the data of the previous survey. Microsoft Excel was used for the data codes.

The interviews were transcribed verbatim and read to prepare for analysis. The average duration of the interviews was 31 min, varying from 14 to 56 min, depending on the participants’ time and experience. To structure and summarize the changes due to the COVID-19 pandemic, content analysis was used, according to the steps of Gioia et al. [47]. Meaningful text segments that explicitly referred to the COVID-19 pandemic were systematically selected, condensed, and paraphrased into first-order codes, that were closely related to the wording of the participants. Subsequently, these first-order codes were grouped into second-order codes, iteratively formed by the main investigator. The second-order codes were translated from Dutch to English. All first- and second-order codes of two participants were checked by another researcher, a native English speaker, the second order codes were recoded, and checked until consensus was achieved. Differences were discussed to improve the accuracy of the codes. The second-order codes were grouped into subcategories and categories to form a data structure that was discussed with two other researchers. Subsequently, the second-order codes were assigned independently into subcategories and categories in a digital workshop by seven other researchers to achieve intercoder agreement. Three of the 53 codes were placed in a different subcategory in the digital workshop compared to the initial data structure. Furthermore, it was suggested to add one extra subcategory, and the names of the categories and subcategories were discussed. These results were used to define the final data structure.

To investigate how the comfort preferences changed, the participants’ preferences from the survey [25] were compared with their preferences from the interviews. Therefore, all relevant text segments, that referred to the ranked IEQ and social comfort aspects were systematically selected. These fragments were condensed and paraphrased in first-order codes and categorized according to predefined categories per participant. This was done for IEQ and social comfort separately. The IEQ-categories were the four IEQ aspects [23], i.e., indoor air quality, thermal comfort, visual quality, acoustics, and logic of ranking IEQ. The social comfort categories were the four most important social comfort aspects that were determined in the survey [25], i.e., contact with colleagues and patients, a safe workplace, sufficient patient privacy, no distraction from noise, and logic of ranking social comfort. The codes were checked and recoded iteratively by two researchers. Then the codes and ranking were compared with the individuals’ preferences of the survey, the changes due to the pandemic, other changes, and the logic of ranking.

To illustrate the IEQ and social comfort experiences [48], quotations by the participants from transcripts verbatim that were detailed and representative, were selected [49]. To justify the unique experiences of the participants [50], gap words and some repetitions of thoughts were kept in the quotations. The quotations were translated back and forward by the authors (native Dutch and English, see Appendix A for the quotations).

## 3. Results

### 3.1. Participants

The interviews were performed in the last week of October and the first half of November in 2020. 130 invitations (38 + 45 + 47) respectively to organizations A, B, and C were sent, 17 (5 + 7 + 5) interviews were included in the analysis, as seen in Figure 4. The main reason for refusing to participate was work pressure. One audio recording was damaged, therefore that participant was excluded from the analysis.

The participants, who consisted of 16 women and one man, represented all IEQ and social comfort clusters and belonged to all the hospital organizations that participated in the survey. The participants worked in consultation rooms, treatment rooms, offices, or at reception desks; thirteen of them worked in more than one room type, see Figure 5. For example, three interviewees worked at reception desks and consultation rooms. 

Nine participants regarded the same IEQ-aspect as most important, as selected in the in the 2019 survey (Table 2). For example, one participant regarded sufficient fresh air as most important in the autumn of 2020, and sufficient fresh air and not too cold or hot important in the spring of 2019. Some found the ranking of IEQ aspects from most important to least important logical, while others did not. No annoyance with noise or sufficient daylight were for some the most important aspects, while the other aspects were equally less important. The main preferences for indoor air quality and thermal quality did not vary, except for one participant who did not answer the question on IEQ-preferences in the survey of 2019. The main preferences for daylight and noise varied.

Ten outpatient workers regarded the same social comfort aspect as most important in both 2019 and 2020. Some outpatient workers found the ranking of social comfort aspects logical, while others perceived an overlap between contact with colleagues and patients and a safe workplace. The importance of social comfort aspects could also depend on a situation or activity. The importance of maintaining the privacy of patients was considered a question of conscience. The main preference for sufficient contact did not vary, while the other preferences did.

### 3.2. Changes Due to the Coronavirus Disease 2019 (COVID-19) Pandemic

The data structure generated insights into organizational aspects, work-related aspects, physical adaptations, and the experience of changes, that were identified by the participants (Table 3). The structure was used to understand the preferences for indoor environmental quality and social comfort during the COVID-19 pandemic. First, the organizational aspects, work-related aspects, and physical adaptations were summarized, to describe the context. Then the experiences, especially related to the indoor environment and social comfort, are explained. See Appendix B, Table A1 for all second-order codes.

#### 3.2.1. Context

In March/April 2020, during the first wave of the pandemic, the capacity of outpatient care had been reduced for most participants. At the time of the interviews, most participants regarded the capacity of outpatient care as normal. The number of patients, relatives, and staff in the hospital buildings was perceived lower than before the COVID-19 pandemic. To limit the number of persons in the hospital buildings, outpatient care had shifted from face-to-face care to digital care. Face-to-face consultations were only performed when necessary or when preferred by patients, also depending on the department. For example, one outpatient worker in an oncology department explained that patients were invited to face-to-face consultations when they were told that they were seriously ill. Diagnostic treatment was performed at the hospital buildings.

The adaptation of work methods resulted in annoyance with extra administration and worries about infection and patient care. Half of the participants perceived increased work pressure at the time of the interviews, while the others did not. Some reception workers had perceived increased work pressure mainly when the COVID-19 pandemic started. Work pressure increased also because of limited staff and prolonged sick leave, not directly related to the COVID-19 pandemic. Positive experiences were easily adjusted to new work methods and satisfaction with the hospital organizations’ diligence.

To support social distancing between persons, some adaptations were executed in the reception areas and adjacent waiting rooms. For example, cupboards were replaced; and stanchions in front of the reception desk were placed (Figure 6). Also, the number of chairs in waiting rooms was reduced, or splash guards were provided between the chairs. Other protective measures were the use of a facial shield or facial mask and a transparent splash guard, placed at reception desks (Figure 6). Both photographs are taken by participants.

#### 3.2.2. Indoor Environmental Quality

The perception of the indoor air quality, thermal comfort, and noise was influenced by the COVID-19 pandemic.

The outpatient workers explained that they preferred to have natural ventilation by opening a window or the door to the corridor, to reduce the risk of infection (For example, quotation 1 in Appendix A, “And I find it annoying too now…”). An adaptive strategy was to open the door to the corridor when the outpatient workers who were in a room without a façade window were dissatisfied with the supply of mechanically ventilated air. This was only possible when it did not affect the privacy of the patients. Another strategy was to open the façade window for natural ventilation. However, this could affect thermal comfort, which might vary between outpatient workers who worked in the same room. When the window was open, it could be too cold for some workers, especially for those in uniform (with bare arms), while it was not too cold for others. An outpatient worker explained that they had clear appointments about control of natural ventilation to achieve a balance of thermal comfort and natural ventilation since the pandemic (quotation 2 in Appendix A, “I like a little bit of air…”).

Also, the speech intelligibility of patients was reduced due to the splash guards between the patient and outpatient workers at the reception desk. The noises from colleagues at the back of the reception area were louder because of the splash guards, while the voice from the patients sounded lower. The facial masks increased difficulties with speech intelligibility (quotation 3 in Appendix A, ”If it weren’t any noise annoyances”).

Furthermore, recent changes in preferences that were not explicitly associated with the COVID-19 pandemic, were experienced. This occurred for noise and daylight. For example, one participant who worked mainly in rooms without a façade window missed daylight more than she used to do. Dry eyes and concentration problems were associated with a lack of daylight. The outpatient worker adapted her behaviour to receive daylight, by going out for a walk during lunchtime and working in rooms with a window when possible (quotation 4 in Appendix A, “I have noticed that I used to suffer less with that…”). These strategies were also explained by some other interviewees, who worked generally in rooms without a façade window.

#### 3.2.3. Social Comfort

The outpatient workers also experienced changes in contact with colleagues and patients, privacy of patients and safety, due to the COVID-19 pandemic.

Some felt less energized because of the reduced face-to-face contact with patients and increased administrational work. The perception of losing information from the patients occurred because of the limited face-to-face contact. For example, one outpatient worker explained that she could not explain doubts to patients anymore after consultation with the physician (quotation 5 in Appendix A, “When you ask…”). Another outpatient worker explained that she was worried about missing details and could give less attention to the patients due to the limited physical examination (quotation 6 in Appendix A, “And yes, the physical check-up is something you miss…”).

A shorter physical distance, which was not allowed due to infection risk, was regarded as advantageous to support the patients’ privacy by speaking softly. The privacy of patients at the reception desk also worsened due to the splash guards and facial masks. For example, the outpatient workers had to ask for personal information, such as the birth date and the name of the general practitioner, while others were in the waiting room. The patients had to talk louder because of the splash guards (quotation 7 in Appendix A, “You discuss everything…”).

Some outpatient workers perceived difficulties through increased aggression. Patients were impatient or angry, for example because of the obligation to wear a facial mask. The opposite was also experienced, because the number of visiting patients had decreased, and aggression during telephone calls was less annoying.

## 4. Discussion

The qualitative data allow for a vivid explanation of the context [38] and changes due to the COVID-19 pandemic. Contextual changes were studied to explain the experience and importance of comfort aspects. Because the participants in this study had been involved in the cluster study [25], a comparison of the preferences was possible. This is presented and discussed in the following section.

### 4.1. Explanation of the IEQ Clusters

Figure 7 shows the preferences for IEQ aspects per cluster, that were found in the survey before the pandemic started, and the preferences during the pandemic. The circles represent the proportion of the outpatient workers who regarded an IEQ aspect important before the pandemic and the ranking of the outpatient workers during the pandemic. The area of the green circles represents the proportion of the cluster members for a main preference. The largest circles (e.g., cluster 1 control of ventilation) represent 100%, the green dots represent 0% of the cluster members. The area of the purple circles represents the ranking of the preferences; large is most important (1, size 100%), small is least important (6, dot). The size of the purple circles was calculated according to the following equation, with *v* = the sum of ranking of the participants per cluster, *n* = number of cluster members:(1)$$100-\left(\left(\frac{v}{n}\right)\times \left(\frac{100}{5}\right)-\left(\frac{100}{5}\right)\right)$$

The number of cluster members from the survey of 2019 and interviews in 2020 are shown as follows: IEQ cluster × (N = a/b): × is the cluster number, a = number of cluster members of the study in 2019, b = number of cluster members of the study in 2020.

The importance of some preferences had a limited variation between both data sets. For example, the proportion of outpatient workers in cluster 1 who found control of temperature important was intermediate (46%), and ranking was intermediate (3.3, of six aspects with 1 for most important, 6 least for important). Other preferences differed; for example, none of the outpatient workers in cluster 1 selected not too cold or hot among the three most important aspects in the quantitative study (0%), while it was ranked intermediately (3.8) in the qualitative follow-up study. The largest differences were not too hot or cold (cluster 1), control of temperature (cluster 4), control of ventilation (cluster 2), sufficient daylight (cluster 3), no annoyance by noise (cluster 5).

These differences suggest that the clusters that were mainly differentiated by the preferences could change due to contextual or personal changes. For example, one of the main contextual changes due to the COVID-19 pandemic was the reduced number of persons in the buildings and rooms. The IEQ-clusters mutually differed for those working in rooms with two to four persons and rooms with more than four persons. It was explained by the outpatient workers of cluster 4 and cluster 5 that the preferences of colleagues were one of the factors that influenced the control of temperature and ventilation.

The reason why the outpatient workers found IEQ aspects important, varied between the clusters. For example, in addition to concerns for infection with the SARS-CoV-2 virus, control of ventilation and sufficient fresh air were preferred because fresh air was experienced as enjoyable and attractive for those in clusters 2 and 6. However, the outpatient workers in cluster 1 found control of ventilation and sufficient fresh air important because they were dissatisfied with the indoor air quality. Those in cluster 1 found thermal comfort important because they were mainly dissatisfied with the temperature variation, while it was important in cluster 4 because of draught. Thermal comfort was important for those in cluster 5 because they mainly experienced too cold temperatures even after adjusted clothing (e.g., cardigan on top of uniform). Daylight was preferred because of dissatisfaction with the absence of a façade window in cluster 1, while some in clusters 4, 5, 6 did not need or expect a façade window. The view to the outside, which was associated with sufficient daylight, was missed (cluster 1) or enjoyed (cluster 2, 3, 6). The importance of noise was for those in clusters 1 and 6 mainly related to speech intelligibility of patients, which had decreased due to wearing of facial masks and splash guards, while outpatient workers in clusters 3 and 5 perceived stress due to annoyance by noise.

Based on the explanation of the data from the survey and interviews, it can be suggested that the needs of those in different IEQ clusters do not only vary in importance, but also due to differences in expectations and sensitivity. The clusters seem to be influential for contextual changes or personal changes, due to large differences, especially in clusters 2, 3, and 4.

### 4.2. Explanation of the Social Comfort Clusters

Figure 8 shows the preferences for social comfort that were found in the survey before the pandemic started, and the preferences during the pandemic. The size of the circles represents the importance of a specific comfort aspect. The area of the green circles represents the proportion of the cluster members for a preference. The green dots represent 0% of the cluster members. The area of the purple circles represents the ranking of the preferences; large is most important (1, size 100%), small is least important (4, dot). The size of the purple circles was according to the following equation, with *v* = the sum of ranking of the participants per cluster, *n* = number of cluster members:(2)$$100-\left(\left(\frac{v}{n}\right)\times \left(\frac{100}{3}\right)-\left(\frac{100}{3}\right)\right)$$

The number of cluster members from the survey of 2019 and interviews in 2020 are shown similarly as in Figure 7. The importance of most preferences for social comfort in 2020 did not vary from 2019. For example, contact with colleagues and patients was in all clusters for a large majority (ranging from 56% to 76%) important in 2019, and ranked similarly high (1.5 to 1.8, of four aspects, with 1 for most important and 4 for least important) in 2020. The largest difference was the limited proportion (0%) of outpatient workers who found a safe workplace important in 2019, while it was ranked intermediately (2.5) in 2020. The reason why those in cluster 1 regarded safety as important was different from those in clusters 2 and 3. In cluster 1 safety was associated with building characteristics, such as a safe escape route or cleanliness, while safety was associated with verbal aggression of patients and the proximity of colleagues in clusters 2 and 3. The reason why the outpatient workers found no distraction by noise important, varied partly. In cluster 1 concentration was necessary, while in all clusters noise from others, particularly telephone calls of colleagues, was perceived as annoying. This can be explained by the results from the survey, that indicated that those in cluster 1 perform generally more concentrated office work in comparison to those in cluster 2 and 3.

The reasons why the preferences for the privacy of patients and contact with others were important, did not vary between the clusters. Patient privacy was mainly important because of concerns about the audibility of personal information by other patients in the waiting room. Contact with colleagues and patients was perceived as important because of work satisfaction. Among the clusters, both collaboration with colleagues and interaction with patients contributed to work satisfaction.

Overall, it can be suggested that the importance of safety can vary between those in different social comfort clusters, while it is less likely that the reasons for the importance of the other social comfort aspects vary. The social comfort clusters seem generally robust, as the changes were limited between 2019 and 2020.

### 4.3. Changes of Preferences

The study shows that it is likely that the occupants’ preferences can change due to contextual changes.

As choices for preferences are inevitably made in a context, it is difficult to distinguish contextual influences from preferences [51]. Preferences can be formed unconsciously by habituation and more consciously by reasoning [52]. The latter are more likely to change due to contextual cues [51,53]. A study that used neuroimaging data, showed that physiological reactions, representing the perceived importance of preferences, can strengthen after selection [54]. Hoeffler and Ariely (1999) suggested that a strong experience is more likely to form a stable preference, compared to a flawed experience [55]. The present study showed that the main preferences for daylight, noise, privacy for patients, and safety were more likely to change than the other IEQ and social comfort aspects. Future study is needed to indicate whether there are differences between comfort aspects in the strength of the experiences.

### 4.4. IEQ in Relation to Changes Due to the COVID-19 Pandemic

The present study shows that the influence and interrelations of physical characteristics, work, and personal aspects with comfort preferences are complex. For example, while the needs for control of natural ventilation and for sufficient fresh air, due to worries about infection risk, were expressed by seven participants, indoor air-related aspects were most important for three participants. Their main preference for indoor air-related aspects had not been altered since 2019. A possible cause is that some outpatient workers found that other IEQ-aspects affected their work performance more negatively if these were not met. The participants pointed out that opening windows influenced also thermal comfort, while opening doors influenced annoyance by noise.

However, the concern of the outpatient staff about indoor air quality as a possible risk factor for health and infection is not new. In previous studies, hospital staff regarded indoor air quality as among the top three most important aspects [26,56]. Furthermore, evidence of the possible transmission of SARS-CoV-2 through (small) airborne particles is growing [57]. For example, a study in the isolation wards of intensive care determined contamination with the SARS-CoV-2 virus on surfaces at the nursing station and in the indoor air [58]. The ventilation rate in the isolation wards was low, while tracheal intubation that may increase the concentration of airborne virus-carrying particles had been performed the day before data collection. Increased ventilation rate is one of the measures that can contribute to a lower concentration of airborne virus-carrying particles. Customization of mechanical ventilation systems and control of air supply through opening windows were among the measures recommended by Morawska et al. (2020) [59], to reduce the risk of the spread of the SARS-CoV-2 virus.

### 4.5. Social Comfort in Relation to Changes Due to the COVID-19 Pandemic

The experience of impoverished interaction, due to increased digital care, can be explained by the multi-sensory characteristics of face-to-face interaction. Similarly, the beneficial effects of face-to-face interaction for collaboration were identified in previous studies [60]. For example, a comparative study on the difference between video and face-to-face meetings of physicians found less informal exchange and limited willingness to discuss diagnostic problems through video calls as compared with face-to-face meetings [61].

There is a gap in empirical studies on the perception by caregivers of digital care during the COVID-19 pandemic [62,63]. However, previous comments on the rapid shift to digital care during the pandemic suggested changes in social interaction through video calls in comparison to face-to-face meetings. For example, Romanick-Schmiedl and Ragu (2020) [64] suggested that face-to-face interaction between patient and caregiver contributed to a trusting relationship, which is essential for the health care process. Furthermore, cues for correct diagnosis of the patient, such as observation of trembling fingers, could be missed in virtual contact. Rosen et al. (2020) [65] suggested that contact with patients might improve through digital care because the patients are comfortable in their own homes. Notwithstanding the benefits of digital care, in terms of infection risk, expenses, travel, and time, the perception of social comfort can be influenced negatively by the shift from face-to-face meetings to digital care.

### 4.6. Methodological Considerations

The lens that was specifically chosen for this study provided insight into changes due to the COVID-19 pandemic and other factors. Other lenses such as the experience of basic and linear factors derived from Kano’s model for satisfaction [66] or different adaptive strategies to achieve comfort [14,15], could have allowed us to analyse the preferences. However, because it was unknown whether the outpatient areas were changed due to the COVID-19 pandemic, it seemed most appropriate to form a data structure of the changes, that the participants related to the pandemic.

In the present study, a ranking was used because this method is most suitable to assess a hierarchy of preferences, while rating scales are most suitable to assess appraisal [67]. One of the benefits of ranking preferences instead of rating them is to overcome differences in the assessment between persons [68]. Therefore, the ranking was also used for the preferences (three most important aspects) and the rating for satisfaction with comfort (scale 1 to 7) in the survey [25]. Subsequently, the clusters were produced with the categorical values for preferences (binary data) and components of comfort (continuous data from principal component analysis) with two-step cluster analysis. This technique is suitable for both types of data [24]. The present study shows that differentiation of preferences can be illogical for outpatient workers because they find some or all comfort aspects evenly important. Furthermore, interrelations were perceived between the comfort aspects. Therefore, future research is needed to compare the consistency of ranking and rating for comfort preferences.

### 4.7. Limitations

One of the limitations of this study can be participation bias. To limit this bias, participants from all clusters, differing in preferences and comfort, were represented in this study. Therefore, it was expected that the participants would constitute a group of outpatient workers with different main preferences and satisfaction with comfort. Although the preferences of some participants had changed since 2019, the main preferences of the total sample differed also in the present study.

Another limitation is that some outpatient workers found the ranking of comfort aspects to be illogical. The perceived lack of logic to rank comfort aspects might have influenced the identification of preferences. For example, participants in cluster 3 regarded differentiation of preferences as illogical, while their preference for daylight changed. Furthermore, careful consideration is needed for the generalization of the results, mainly of IEQ clusters 2, 3, 4, because of the low number of participants. Also, transfer of the findings to other occupant groups (such as patients) or departments (such as inpatient areas) needs careful consideration, because the occupant needs can vary due to differences in building characteristics, duration of stay, activities, etc. [20,21].

### 4.8. Recommendations

This study shows that the main preferences of the outpatient workers can vary. Based on the results it can be suggested that manual control of IEQ aspects is one of the solutions to improve the comfort of individuals. A previous study on thermal comfort of hospital workers at inpatient areas recommended hospital organizations accommodate different set points, related to zones that varied in occupancy and activities [69]. However, as the preferences of individuals that work together in the same area can vary, additional solutions are needed. Other solutions that hospital organizations can accommodate for are e.g., adjusted clothing (uniforms), use of other rooms, or compensation during breaks. In line with compensation during breaks, Lembo et al. (2021) suggested reducing the duration of work shifts during the COVID-19 pandemic because of thermal discomfort of the hospital workers that used personal protective equipment [70]. Also, measures to increase the ventilation rate may improve comfort and reduce worries of outpatient workers. Ventilation could be improved and reduce the risk of the spread of the SARS-CoV-2 virus through the opening of windows, air filtration, disinfection, and accurate operation of the HVAC systems [59,71]. Jain et al. (2021) [72] addressed the importance of a correct balance between the occupants’ comfort and energy use of HVAC systems in hospitals. They suggested developing strategies for measurement and control of IEQ including measurement of the system performance. Furthermore, as the intelligibility of patients may decrease due to splash guards and facial masks, additional acoustic measures are needed during an epidemic. Possible solutions are reconsideration of splash guards, application of extra sound absorbing materials and reduction of environmental noises. This is important because a poor acoustic environment and reduced privacy may increase the incidence of burn out of healthcare workers [73], a risk that increased during the COVID-19 pandemic [74]. Furthermore, it can be suggested that places that accommodate safely for face-to-face contact with hospital workers and patients are needed for medical and informal informal exchange. Places for interaction with others may also decrease the healthcare workers’ work pressure [75]. To support social distancing and face-to-face interaction, the occupant density of rooms, areas, and corridors might be considered [71].

## 5. Conclusions

In this study, the preferences for IEQ and social comfort of the outpatient workers during the COVID-19 pandemic were investigated and compared to preferences for IEQ and social comfort identified before the pandemic started. The perceived changes of adaptations to reduce the infection risk of the SARS-CoV2 virus in hospitals were summarized. The outpatient workers had worries about the indoor air quality, were annoyed by a decreased speech intelligibility with patients, impoverished interaction, increased difficulties with patient privacy, and by their threatening behaviour. The study allowed us to compare preferences for IEQ and for social comfort with the interviewees’ preferences that were identified in a survey before the COVID-19 pandemic started. The results from the previous study identified six clusters for IEQ and three clusters for social comfort that were distinguished by their preferences and comfort. The present study showed that the reason why IEQ aspects are important varied between the clusters, while the variation for social comfort aspects was limited.

For some of the outpatient workers, differentiation was illogical due to interrelations and the equal importance of the comfort aspects. This was the case with IEQ as well as with social comfort.

Finally, the study implies that the occupants’ preferences for IEQ and social comfort can change over time, due to contextual or personal changes. Therefore, it can be suggested that further development of occupant profiles that might be used in the programmatic or design phase of renovation and newly built outpatient areas is needed.

## Figures and Tables

**Figure 1 ijerph-18-07353-f001:**
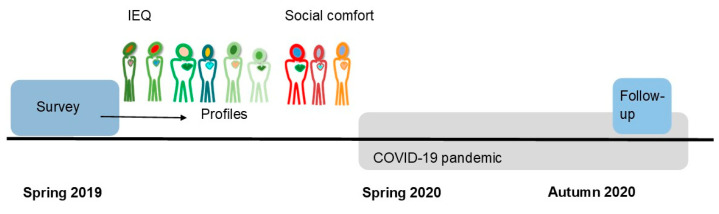
Timeline profiles.

**Figure 2 ijerph-18-07353-f002:**
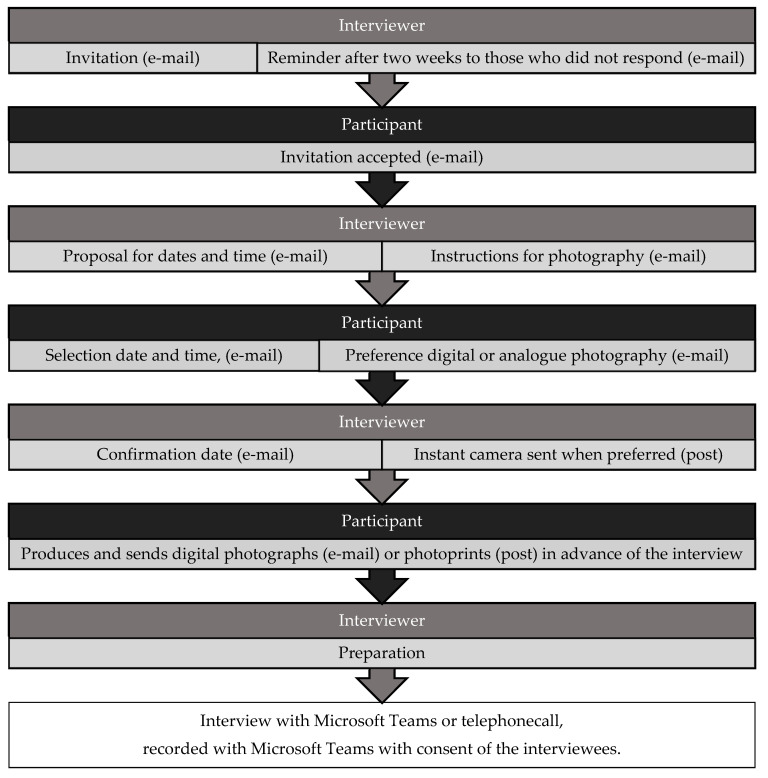
Procedure.

**Figure 3 ijerph-18-07353-f003:**
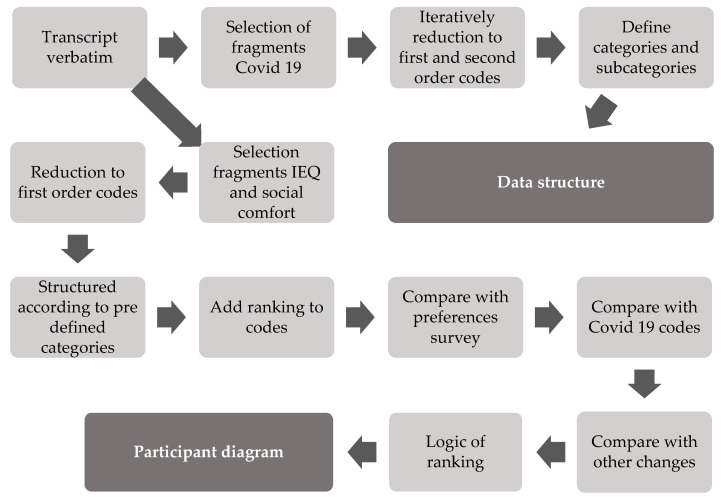
Steps for data analysis.

**Figure 4 ijerph-18-07353-f004:**
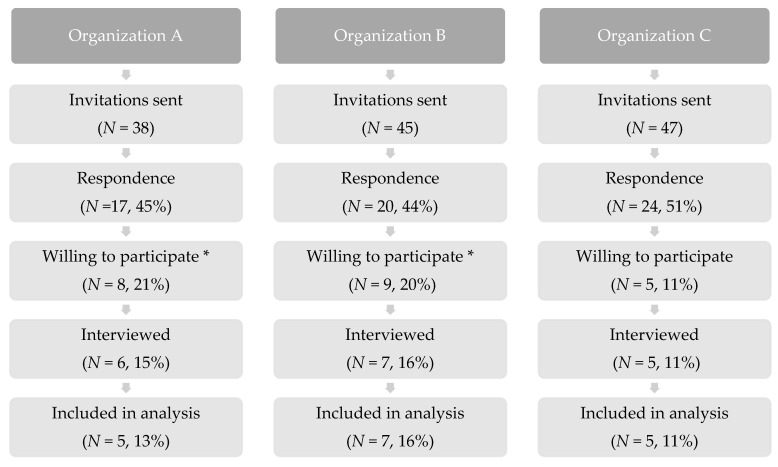
Flow diagram of recruitment. * inclusive participants that were not within the inclusion criteria or canceled.

**Figure 5 ijerph-18-07353-f005:**
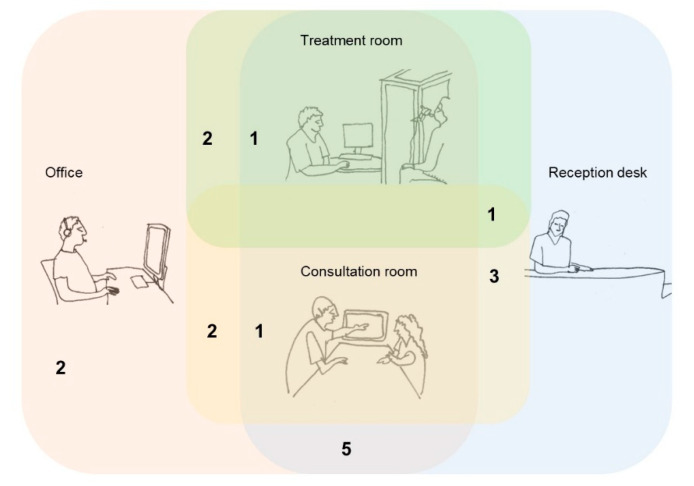
Overlap of workplaces of the participants.

**Figure 6 ijerph-18-07353-f006:**
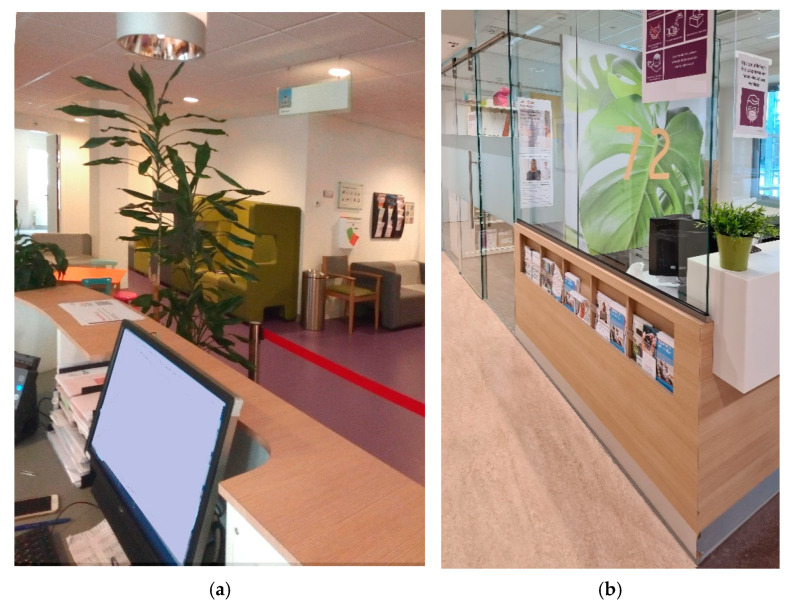
(**a**) Stanchions in front of reception desk; (**b**) splash guard installed at reception desk.

**Figure 7 ijerph-18-07353-f007:**
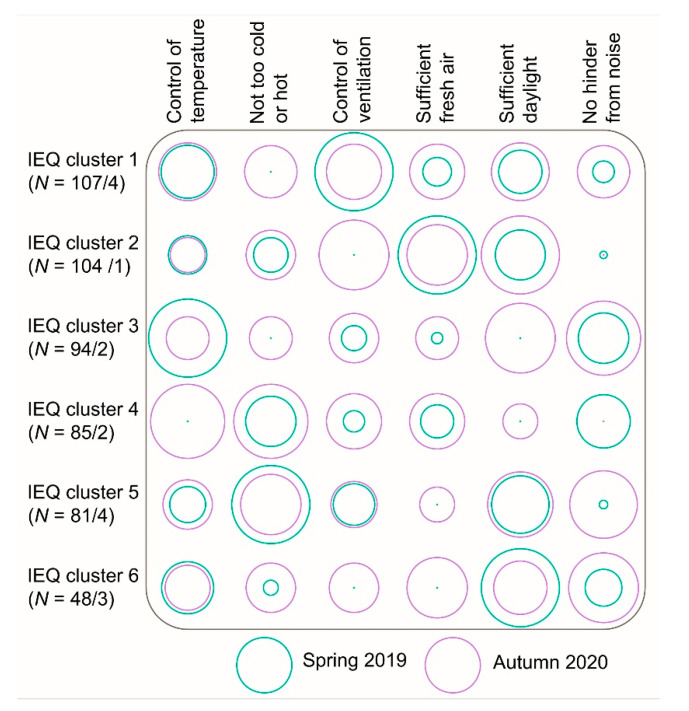
Proportion of preferences selected per IEQ cluster in 2019 and average rating of preferences in 2020.

**Figure 8 ijerph-18-07353-f008:**
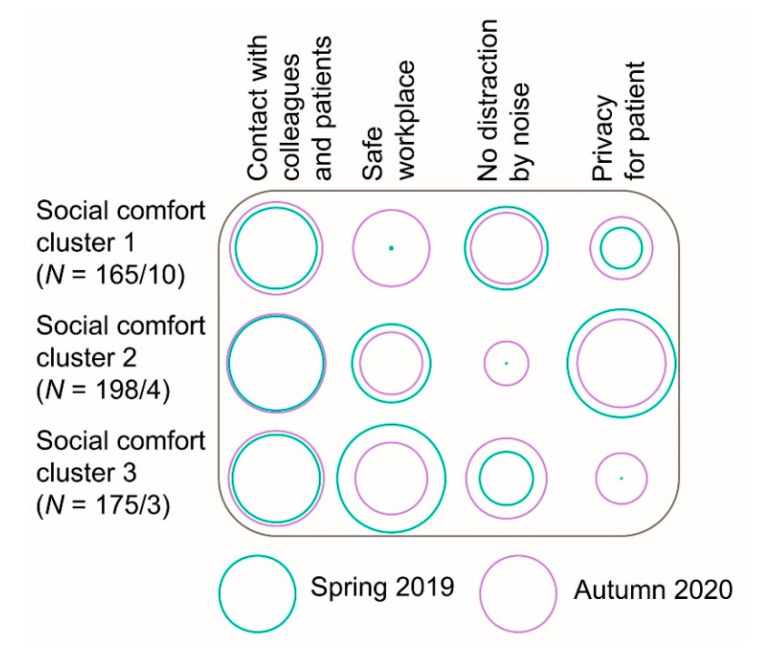
Proportion of preferences selected per social comfort cluster in 2019 and average rating of preferences in 2020.

**Table 1 ijerph-18-07353-t001:** Topic guide semi-structured interviews.

Main Topic	Subtopics	Shared Documents
Introduction		
Work-related aspects	Room, room type, number of persons in room, function, job tasks, department, location	
Changes due to the pandemic	Activities, work pressure, physical changes, other	
Preferences IEQ	Thermal, ventilation, lighting, noise	Photographs
Preferences social comfort	Contact, safety, privacy, distraction	Photographs
Ranking preferences	Order of importance Logic to distinguish preferences	List of ranking IEQ and social comfort
Other issues related to comfort		
Closing		

**Table 2 ijerph-18-07353-t002:** Main preference in 2020 compared to 2019.

	Most Important Aspect	2020	Same Preference in 2019
		N	N
Indoor Environmental Quality (IEQ) IEQ	Sufficient fresh air	1	1
	Control of ventilation	2	2
	Not too cold or hot	3	2 *
	Control of temperature	1	1
	Sufficient daylight	5	1
	No annoyance by noise	5	2
Total IEQ		17	9
Social comfort	Sufficient contact	7	7
	No distraction by noise	3	2
	Privacy for patients	4	1
	Safe workplace	3	0
Total Social comfort		17	10

* Incomplete answer of participant in 2019 excluded.

**Table 3 ijerph-18-07353-t003:** Data structure of the changes related to the coronavirus disease 2019 (COVID-19) pandemic.

Category	Subcategory	Example
Organizational adaptations	Capacity	Reduction capacity during first wave of the pandemic
	Number of persons in hospital building	Working partly from home
Adaptations work methods	Corona care	Perform corona tests in triage tent
	Digital care	Prepare digital consultations
	Face-to-face care	Physical examination when urgent
Physical adaptations	Room	Stanchions in front of reception desk
	Protective clothing	Use of facial mask
Personal experiences	Work-pressure	Increased work pressure
	Satisfaction with work	Annoyed by extra administration
	Indoor Environmental Quality (IEQ)	Missing control of ventilation
	Social comfort	Missing face-to-face contact

## Data Availability

More information on the data presented in this study is available on request from the corresponding author. The data are not publicly available due to restrictions regarding the privacy of the participants.

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
