# Peer review of "Preferences for Indoor Environmental and Social Comfort of Outpatient Staff during the COVID-19 Pandemic, an Explanatory Study"

_ijerph, 2021, doi:10.3390/ijerph18147353_

Round 1

Reviewer 1 Report

Dear authors, thank you for the opportunity to see your work!
Undoubtedly, this aspect of the problem is very important.
At the same time, there are a number of comments on your work to improve understanding and meaning for readers:
1. It is necessary to supplement the introduction with an additional review of sources devoted to the study of the social comfort of medical personnel in various conditions of activity.
2. More clearly formulate the hypotheses of the research and indicate their rationale.
3. It is necessary to expand the section "Discussion of results": correlate the results of your research with a large number of studies of other authors, and also write which authors see practical recommendations for management and the employees themselves, made on the basis of the results of this study.

Author Response

Reviewer: 1

Dear authors, thank you for the opportunity to see your work!
Undoubtedly, this aspect of the problem is very important.
At the same time, there are a number of comments on your work to improve understanding and meaning for readers:
1. It is necessary to supplement the introduction with an additional review of sources devoted to the study of the social comfort of medical personnel in various conditions of activity.

Response

We thank the reviewer kindly for reviewing the manuscript and the comments. We are very grateful for the suggestions because these contributed to improvement of the manuscript. Additional examples on social comfort of hospital workers are added as follows (page 2, line 43):

“It has been suggested by several authors that IEQ as well as social comfort aspects are important to understand health and comfort [14-16]. Privacy and interaction have been included in previous studies. For example, nurses that moved from open bay wards to a ward with 100% single patient rooms, missed the informal interaction with colleagues in the new wards [17]. Also, the exchange of medical information was better at open wards, that comprised of 36 beds without separation walls, than at bay wards with walls between 4-6 beds [18]. Another example is the relation of the type of communication (case related or comforting) with room types at an Emergency Department [19].“

  1. More clearly formulate the hypotheses of the research and indicate their rationale.

Response

Thank you for the comment. The hypotheses and rationale were more clearly formulated as follows (page 2, line 94):

“Because of limited information on the hospital workers’ preferences for comfort, this The study that is presented in this paper, aims to explain the differences in preferences of outpatient workers, that were identified by in the clusters in of the 2019 survey in 2019. This study provides insights into the comfort experiences of outpatient workers during the COVID-19 pandemic and changes of their preferences during the pandemic.”

  1. It is necessary to expand the section "Discussion of results": correlate the results of your research with a large number of studies of other authors, and also write which authors see practical recommendations for management and the employees themselves, made on the basis of the results of this study.

Response

Thank you for the suggestion. We added practical recommendations based on the results of this study, including studies of others, as follows (page 16 line 568):

“4.7 Recommendations

This study shows that the main preferences of the outpatient workers can vary. Based on the results it can be suggested that manual control of IEQ aspects is one of solutions to improve comfort of individuals. A previous study on thermal comfort of hospital workers at inpatient areas recommended hospital organizations to accommodate for different set points, related to zones that varied in occupancy and activities [69]. However, as the preferences of individuals that work together in the same area can vary, additional solutions are needed. Other solutions that hospital organizations can accommodate for are e.g., adjusted clothing (uniforms), use of other rooms, or compensation during breaks. In line with compensation during breaks, Lembo et al. (2021) suggested to reduce the duration of work shifts during the COVID-19 pandemic because of thermal discomfort of the hospital workers that used personal protective equipment [70]. Also, measures to increase the ventilation rate may improve comfort and reduce worries of outpatient workers. Ventilation could be improved and reduce the risk for the spread of the SARS-CoV-2 virus through the opening of windows, air filtration, disinfection, and accurate operation of the HVAC systems [59,71]. Jain et al. (2021) [72] addressed the importance of a correct balance between the occupants’ comfort and energy use of HVAC systems in hospitals. They suggested to develop strategies for measurement and control of IEQ including measurement of the system performance. Furthermore, as the intelligibility of patients may decrease due to splash guards and facial masks, additional acoustic measures are needed during an epidemic. Possible solutions are reconsideration of splash guards, application of extra sound absorbing materials and reduction of environmental noises. This is important because a poor acoustic environment and reduced privacy may increase the incidence of burn out of healthcare workers [73]. A risk that increased during the COVID-19 pandemic [74]. Furthermore, it can be suggested that places that accommodate safely for face-to-face contact with hospital workers and patients are needed for medical and informal information exchanges. Places for interaction with others may also decrease the healthcare workers’ work pressure [75]. To support social distancing and face-to-face interaction, the occupant density of rooms, areas, and corridors might be considered [71].”

Reviewer 2 Report

Some advices:

1) It is nor clear for the reader how the researchers match individual interviews with the participation in the questionnaire developed in 2019: does this initial questionnaire allowed clear identification of participants... it should be explained clearly in the article. Somewhere in the article we can read that "Data security was assessed by a Data manager... To respect privacy of the participants, persons are not traceable from the results presented" and this leads to confusion. Therefore my advice is for the authors to clearly explain how they were able to compare answers (1st and 2nd moment, being those questionnaire and personal interview").

This research is actual, important and crucial to give insights on how to act in hospitals in order to make workers and patients feel safe... therefore impacting strongly in manegement decisions.

Author Response

Some advices:

  1. It is nor clear for the reader how the researchers match individual interviews with the participation in the questionnaire developed in 2019: does this initial questionnaire allowed clear identification of participants... it should be explained clearly in the article. Somewhere in the article we can read that "Data security was assessed by a Data manager... To respect privacy of the participants, persons are not traceable from the results presented" and this leads to confusion. Therefore my advice is for the authors to clearly explain how they were able to compare answers (1st and 2nd moment, being those questionnaire and personal interview").

Response

We thank the reviewer kindly for reviewing the manuscript and the comments. We are very grateful for the advices because these contributed to improvement of the manuscript. We explained how answers were compared as follows (page 6, line 193)

“ If the participants had shared their e-mail address in the previous survey, it was separated from the dataset and secured in a separate document. Comparison of the individuals’ data between the survey and this follow-up study was enabled by a unique number that was assigned to each participant. To respect the privacy of the participants, persons are not traceable from the results presented.”

And as follows on page 6, line 208:

‘’Then the codes and ranking were compared with the individuals’ preferences of the survey, changes due to the pandemic, other changes, and logic of ranking.”

This research is actual, important and crucial to give insights on how to act in hospitals in order to make workers and patients feel safe... therefore impacting strongly in manegement decisions.

Reviewer 3 Report

Thank you for the opportunity of reviewing this manuscript.

This qualitative study used semi-structured interview and photo-elicitation to explain the outpatient workers’ preferences for comfort during the COVID-19 pandemic. The findings provided insight for future design outpatient areas to accommodate the occupants’ preferences for IEQ and social comfort.

The manuscript was thoroughly prepared with essential information.

I only have some minor comments on data analysis and results as follows:

  1. The authors mentioned that “first-order codes were grouped into second-order codes, iteratively formed by the main investigator. All first- and second-order codes of two participants were checked by another researcher, a native English speaker, re-coded, and checked again”.

How accurate in interpretation of the meaning of the first-order codes should be explained to enhance the credibility of the data.

  1. It would be better to present the results with direct quotes from participants so that the readers can make sense of the coding and enhance the vividness of data.

Author Response

Thank you for the opportunity of reviewing this manuscript.This qualitative study used semi-structured interview and photo-elicitation to explain the outpatient workers’ preferences for comfort during the COVID-19 pandemic. The findings provided insight for future design outpatient areas to accommodate the occupants’ preferences for IEQ and social comfort.

The manuscript was thoroughly prepared with essential information.

I only have some minor comments on data analysis and results as follows:

  1. The authors mentioned that “first-order codes were grouped into second-order codes, iteratively formed by the main investigator. All first- and second-order codes of two participants were checked by another researcher, a native English speaker, re-coded, and checked again”. How accurate in interpretation of the meaning of the first-order codes should be explained to enhance the credibility of the data.

Response

We thank the reviewer kindly for reviewing the manuscript and the comments. We are very grateful for the comments because these contributed to improvement of the manuscript. We added an explanation about the interpretation of the meaning as follows (on page 6, line 215)

“All first- and second-order codes of two participants were checked by another researcher, a native English speaker, the second-order codes were recoded, and checked again until consensus was achieved. Differences were discussed to improve the accuracy of the codes.”

  1. It would be better to present the results with direct quotes from participants so that the readers can make sense of the coding and enhance the vividness of data.

Response

Thank you for this suggestion. We have been considering to present the results with direct quotes before, but had a concern that the manuscript would become too lengthy. To overcome this problem we quoted the first few words and referred to the appendix for the complete quotes in English and Dutch. This was done as follows:

Page 11, line 323

“(For example, quotation 1 in Appendix A, “I like a little bit of air...”)”

Page 11, line 333

“(quotation 2 in Appendix A, “And I find it annoying too now...”)”

Page 11, line 338

“(quotation 3 in Appendix A, “If it weren’t any noise annoyances...”)”

Page 11, line 346

“(quotation 4 in Appendix A, “I have noticed that I used to suffer less with that...”)”

Page 11, line 356

“(quotation 5 in Appendix A, “When you ask...”)”

Page 11, line 359

“(quotation 6 in Appendix A, “And yes, the physical check-up is something you miss...”)”

Page 11, line 365

“(quotation 7 in Appendix A, “You discuss everything...”)”

Reviewer 4 Report

I appreciate the opportunity to review this manuscript and hope my comments assist in the revision process. The material is interesting and the topic is relevant. The method seems to have been followed faithfully and the authors were well-positioned to conduct the analysis. Despite these positives in my view the paper needs more work before it could be published and I have made some specific suggestions below.

- The literature addressed is not described accurately so far as I can see. Relevant literature should be presented more deeply in order to support the research problem. Further, there is no clear distinction between manuscript sections in terms of the content they report. First, I suggest dividing the section "Background" into three components, respectively introduction (explain the general argument of the paper, without going into specific details) background (situate the study concepts within the context of extant knowledge, discuss the international relevance of the concepts) and purpose, creating greater clarity in the analysis of the reader. What is the study's biggest contribution? The contribution should be clearly stated in the introduction.

Data collection

- There are no sources to help the reader understand the qualitative approach and the research paradigm taken in the paper. This needs to be expanded, clarified, and supported by in-text citations.

- The ethical aspects in collecting data are not specifically clarified, independently of the voluntary nature of the subjects´ participation and the approval by the local IR; variables such as the offer of incentives to participate (how participants were compensated for participation), sharing and use of data are not patent. More information is needed about the issues around informed consent or confidentiality or how they have handled the effects of the study on the participants during and after the study.

- Interviews: How were the questions chosen? What was the process? The interview guide was developed based on instruments previously used in other studies?

Discussion

- No results should be presented in the discussion section. Please reorganize the results and discussion sections.

- How valuable is the research? Do they consider the findings in relation to current practice or policy or relevant research-based literature?

- How were themes, concepts and categories generated from the data? Whether analysis was computer-assisted (and, if so, how). There are some conclusions drawn which have neither literature review nor research to support them. It is also not entirely clear whether this is deductive or inductive logic employed here.

- The recommendations/implications for practice/research/education/management should have been approached in greater depth. Please discuss whether or how the findings can be transferred to other populations.

CHECKLIST FOR STYLE

- The manuscript will serve a broad audience of students, researchers, and practitioners, however, the manuscript needs to be carefully and attentively proofread, because some sentences are awkwardly constructed, punctuation is deficient, and therefore reading is occasionally difficult to follow. Would recommend a thorough technical edit of this paper.

Author Response

We thank the reviewer kindly for reviewing the manuscript and the comments. The response can be found in the attachment.

Round 2

Reviewer 4 Report

Thank you for the opportunity to contribute to the improvement of your manuscript. Overall, your manuscript has improved with the recommendations of the various reviewers. Congratulations.